# Speaking Style Conversion in the Waveform Domain Using Discrete Self-Supervised Units

**Gallil Maimon, Yossi Adi**
School of Computer Science and Engineering
The Hebrew University of Jerusalem
gallil.maimon@mail.huji.ac.il

## Abstract

We introduce DISSC, a novel, lightweight method that converts the rhythm, pitch contour and timbre of a recording to a target speaker in a *textless* manner. Unlike DISSC, most voice conversion (VC) methods focus primarily on timbre, and ignore people's unique speaking style (prosody). The proposed approach uses a pretrained, self-supervised model for encoding speech to discrete units, which makes it simple, effective, and fast to train. All conversion modules are only trained on reconstruction like tasks, thus suitable for any-to-many VC with no paired data. We introduce a suite of quantitative and qualitative evaluation metrics for this setup, and empirically demonstrate that DISSC significantly outperforms the evaluated baselines. Code and samples are available at https://pages.cs.huji.ac.il/adiyoss-lab/dissc/.

## 1 Introduction

Imagine hearing a famous catchphrase of your favourite television character spoken in their voice, but uncharacteristically fast, while atypically emphasising the end. This would immediately raise suspicion that something is "wrong". As humans we learn to recognise familiar people and voices, not only by their voice texture (timbre), but also by their typical speaking style (Williams, 1965). Therefore, a true VC method should convert both voice texture and speaking style (rhythm, F0, etc.). Figure 1 describes this visually.

Traditional VC methods mainly focused on changing the timbre of a given speaker while leaving the speaking style unchanged (Stylianou et al., 1998; Kain and Macon, 1998; Nakashika et al., 2013; Chou et al., 2019; Huang et al., 2021). Recent methods propose to additionally convert speaking style (Qian et al., 2020; Chen and Duan, 2022; Qian et al., 2021; Kuhlmann et al., 2022). However, these mainly use only a single target utterance which does not fully capture speaker prosody.

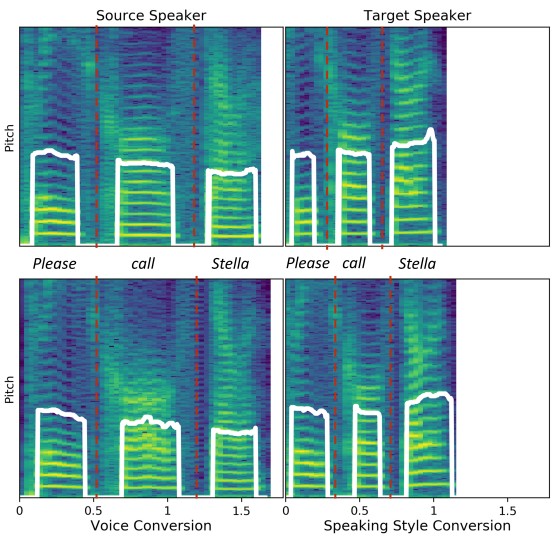

Figure 1: Comparing traditional VC to speaking style conversion. White lines on the spectrograms show the pitch contour. While VC methods change the spectral features so the new utterance sounds like the new speaker, they do not change the rhythm and pitch. Conversely, SSC matches the faster speaking style, and the target speaker's tendency to finish sentences with a pitch increase. This a real sample converted by DISSC. Due to variance within speaking style we do not expect the converted speech to exactly match the target.

Another line of work uses text transcription supervision and shows the potential benefit to VC (Kim et al., 2022; Liu et al., 2021). However, this limits the application to high resource languages and requires large scale data labelling.

Some recent prosody aware VC methods use spectrogram representations as input and output (Qu et al., 2022; Chen et al., 2022), rather than operating in the waveform domain. Thus, they involve another phase of converting from the spectral domain to the time domain using a vocoder. Moreover, using discrete self-supervised speech representations and generating waveforms from these was demonstrated to provide superior performance on plenty of downstream tasks such as

speech and audio language modelling (Lakhotia et al., 2021; Borsos et al., 2022; Qian et al., 2022), multi-stream processing (Kharitonov et al., 2022b), speech emotion conversion (Kreuk et al., 2021), spoken dialogue (Nguyen et al., 2022), speech-to-speech translation (Lee et al., 2022a,b; Popuri et al., 2022), and audio generation (Kreuk et al., 2022a,b).

Inspired by this line of work, we propose a simple and effective method for Speaking Style Conversion (SSC) based on discrete self-supervised and partially disentangled speech representations. Curating samples of different speakers pronouncing the same utterances is challenging, thus we follow the setup of unpaired data, similarly to Lee et al. (2021); Yuan et al. (2021); Kameoka et al. (2018); Kaneko and Kameoka (2017). We give exact details of our approach in Section 3. We formalise the setup of SSC and introduce an extensive evaluation suite in Section 4. Results suggest the proposed approach is greatly superior to the evaluated baselines (see Section 5), while utilising a simple and effective method which can be trained on a single GPU within a couple of hours.

**Our contributions**: (i) We introduce a novel, simple, and effective method which uses discrete self-supervised speech representations to perform textless speaking style conversion in the waveform domain; (ii) We formalise the task of speaking style conversion and propose diverse evaluation methods to analyze different speech characteristics; (iii) We empirically demonstrate the efficacy of the proposed approach using extensive empirical evaluation on several benchmarks considering both objective and subjective metrics.

## 2 Related Work

**Unpaired Voice Conversion.** Many existing methods perform VC in the setting of unpaired utterances (Kameoka et al., 2018; Kaneko and Kameoka, 2017). Most of them contain a vocoder which generates the audio (or an intermediate representation) from several representations, one meant to capture speaker information and others to capture the remaining information (Lin et al., 2021b,a). They encourage disentanglement through information bottlenecks by neural network architecture (Chen et al., 2021; Qian et al., 2019), mutual information and adversarial losses (Lee et al., 2021; Yuan et al., 2021), pretrained models (Huang et al., 2022; Polyak et al., 2021) or combinations of these. Notably, Polyak et al. (2021) used discrete

HuBERT (Hsu et al., 2021) tokens, pitch representation based on YAAPT (Kasi and Zahorian, 2002) and a learned speaker representation to re-synthesise the audio waveform. They showed how replacing the speaker representation at inference time performs VC. Inspired by this modeling paradigm, our approach additionally introduces speaking style modeling and conversion (i.e., pitch contour and rhythm) thus performing SSC.

**Speaking Style Conversion.** Various existing methods aim to change the prosody of a given utterance. However, many convert it based on a single target utterance (Qian et al., 2020; Chen and Duan, 2022) which does not truly capture the general speaking style of the speaker, and often creates artifacts when the contents do not match. Other methods only linearly alter the speaking rate (Kuhlmann et al., 2022) thus ignoring the change of rhythm for different content.

Even leading SSC methods are still limited. Some focus only on rhythm, paying less attention to pitch changes (Qian et al., 2021; Lee et al., 2022c). Other approaches require text transcriptions for training (Lee et al., 2022c; Liu et al., 2021). In contrast, our approach uses HuBERT units trained in a SSL fashion without using textual annotations, thus can support lower resource languages (see Fig. 6 in Appendix A.5). In addition, two concurrent methods for SSC (Qu et al., 2022; Chen et al., 2022) were proposed recently. Although showing impressive results, they are based on spectrogram representations, hence require an additional vocoding step.

## 3 Our Approach - DISSC

We operate under the following setup: we perform *any-to-many* SSC which means we take any recording and convert it to any of the training set speakers. Our training set contains only *unpaired utterances* - e.g. no training utterance is said by more than one speaker. The learning process is completely *textless* with no form of text supervision.

Our approach uses **DI**screte units for **S**peaking **S**tyle **C**onversion, and is denoted DISSC. We decompose speech to few representations to synthesise speech in the target speaking style. We consider three components in the decomposition: phonetic-content, prosodic features (i.e., F0 and duration) and speaker identity, denoted by $\mathbf{z}_c, (\mathbf{z}_{dur}, \mathbf{z}_{F_0}), \mathbf{z}_{spk}$ respectively. We propose a cascaded pipeline: (i) extract $\mathbf{z}_c$ from the raw wave-

form using a SSL model; (ii) predict the prosodic features of the target speaker from $\mathbf{z}_c$ and $\mathbf{z}_{spk}$; (iii) synthesise the waveform speech from the content, predicted prosody and target speaker identity. See Figure 2 for a visual description of DISSC. This modelling approach means moving from a continuous space to a discrete space, hence resulting in easier optimisation and better quality generations.

## 3.1 Speech Input Representation

**Phonetic-content representation.** To represent speech phonetic-content we extract a discrete representation of the audio signal using a pre-trained SSL model, namely HuBERT (Hsu et al., 2021). We use a SSL representation for phonetic-content, and not text or text based methods, to maintain non-textual cues like laughter and allow support for diverse languages. We discretise this representation to use as a rhythm proxy (by number of repetitions) and for ease of modelling. We chose HuBERT for the phonetic-content units as Polyak et al. (2021) showed it better disentangles between speech content and both speaker and prosody compared to other SSL-based models.

Denote the domain of audio samples by $\mathcal{X} \subset \mathbb{R}$. Audio waveforms are therefore represented by a sequence of samples $\mathbf{x} = (x^1, \dots, x^T)$, where each $x^t \in \mathcal{X}$ for all $1 \leq t \leq T$. The content encoder $E_c$ is a HuBERT model pre-trained on the LibriSpeech corpus (Panayotov et al., 2015). The input to the content encoder $E_c$ is an audio waveform $\mathbf{x}$, and the output is a latent representation with lower temporal resolution $\mathbf{z}' = (z_c'^1, \dots, z_c'^L)$ where $L = T/n$. Since HuBERT outputs continuous representations, one needs an additional k-means step in order to quantise these representations into a discrete unit sequence. This sequence is denoted by $\mathbf{z}_c = (z_c^1, \dots, z_c^L)$ where $z_c^i \in \{1, \dots, K\}$ and $K$ is the vocabulary size. For the rest of the paper, we refer to these discrete representations as "units". We extracted representations from the final layer of HuBERT and set $K = 100$. When we wish to predict the rhythm, we must decompose it from the sequence, therefore repeated units are omitted (e.g., $0, 0, 0, 1, 2, 2 \rightarrow 0, 1, 2$) and the number of repetitions correlates to the rhythm. Such sequences are denoted as "deduped".

**Speaker representation.** Our goal is to convert the speaking style while keeping the content fixed. To that end, we construct a speaker representation $\mathbf{z}_{spk}$, and include it as additional conditioning dur-

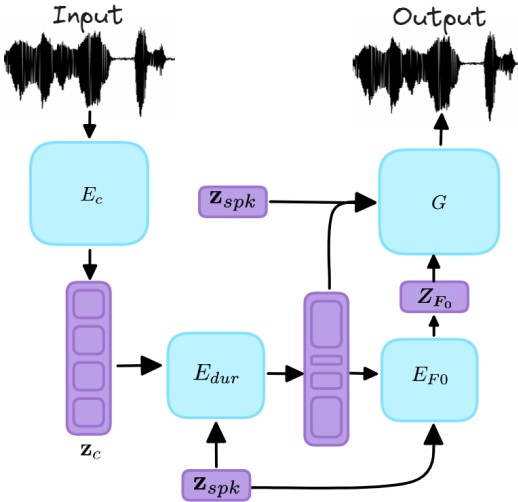

Figure 2: An overview of DISSC. We use an SSL pre-trained content encoder $E_c$ to extract discrete units from the waveform. We independently train a rhythm predictor $E_{dur}$ and pitch predictor $E_{F_0}$ to reconstruct the original unit repetitions and pitch contour, respectively. We input all representations into a pretrained vocoder $G$.

ing the prosody prediction and waveform synthesis phases. To learn $\mathbf{z}_{spk}$ we optimise the parameters of a fixed size look-up-table. Although such modeling limits our ability to generate voices of new and unseen speakers, it produces higher quality generations (Polyak et al., 2021).

**Prosody representation.** As explained in the content representation section, the number of repetitions of units indicates the rhythm of the speaker in the context. More repetitions means a sound was produced for a longer time. Therefore when converting rhythm the sequence is "deduped", and when only converting the pitch the full $\mathbf{z}_c$ is used.

Following Polyak et al. (2021) we encode the pitch contour of a sample using YAAPT (Kasi and Zahorian, 2002). We mark this pitch contour $\mathbf{z}_{F_0} = (z_{F_0}^1, \dots, z_{F_0}^L)$, where $z_{F_0}^t \in \mathcal{R}_{\geq 0}$. We mark the per-speaker normalised version as $\widetilde{\mathbf{z}}_{F_0}$.

## 3.2 Speaking Style Conversion

Using the above representations we propose to synthesise the speech signal in the target speaking style. We aim to predict the prosodic features (duration and F0) based on the phonetic-content units, while conditioned on the target speaker representation. We inflate the sequence according to the predicted durations, and use it, the predicted pitch contour, and the speaker vector as a conditioning for the waveform synthesis phase.

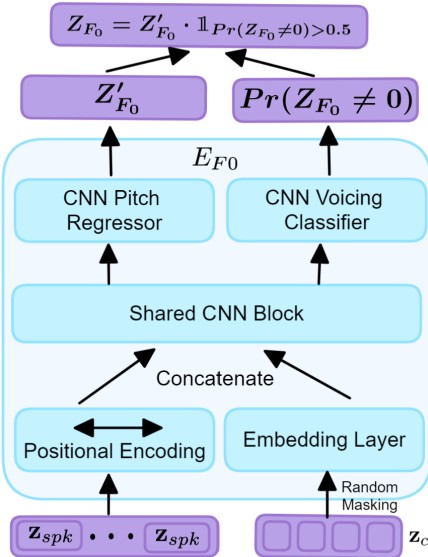

$$Z_{F_0} = Z'_{F_0} \cdot \mathbb{1}_{Pr(Z_{F_0} \neq 0) > 0.5}$$

$$Z'_{F_0} \qquad Pr(Z_{F_0} \neq 0)$$

$E_{F0}$

CNN Pitch Regressor

CNN Voicing Classifier

Shared CNN Block

Concatenate

Positional Encoding

Embedding Layer

Random Masking

$\mathbf{z}_{spk} \quad \cdots \quad \mathbf{z}_{spk}$

$\mathbf{z}_c$

Figure 3: An overview of the $E_{F_0}$ architecture. The positional encoding is a linear positional embedding relative to the start and the end of the sequence, added to the repeated speaker vector. The model jointly predicts the presence of vocalised units (binary classification) together with continuous F0 values for the voiced units.

**Rhythm prediction.** Each HuBERT unit correlates with the phoneme in the given 20 ms. This means that longer repetitions of the same unit indicate a longer voicing of the sound - i.e. rhythm. In order to learn to predict, and convert the rhythm we build a model which learns to predict unit durations, denoted as $E_{dur}$. During training of $E_{dur}$, we input the deduped phonetic-content units $\mathbf{z}_c$ and speaker representation conditioning $\mathbf{z}_{spk}$, with the original unit durations (before dedup) as supervision.

This simple reconstruction task is formulated as a regression task with Mean Squared Error (MSE) loss. As the units do not perfectly correlate to phonemes, neighbouring units often represent the same phoneme even after "dedup". This makes it harder to predict each unit's length separately. Therefore, we encourage the model's nearby errors to cancel each other out by adding an MSE loss on the total error of 4 neighbouring units. This helped with errors all being on the negative end and creating a shortening bias in our preliminary test. In addition, we round the regression outputs to be used as integer repetitions of the unit, however this can also cause a bias in which predictions like $1.51, 1.51, \ldots, 1.51$ and $2.49, 2.49, \ldots, 2.49$ - would result in the same total length. To mitigate this we carryover the remainder after rounding and sum those differences.

During conversion we replace the speaker vector with that of the target and wish for the model to convert the rhythm. To this end our predictor model must strongly depend on $Z_{spk}$. In order to encourage this we introduce two minor adjustments. HuBERT units hold speaker information and are not fully disentangled, however, the shorter the sequence the less information exists (Kharitonov et al., 2022a). We therefore use a CNN with a small receptive field. In addition, we introduce train time masking of units in the sequence, this means that $E_{dur}$ has access to even less units at train time, hence they contain less speaker information. Exact details can be found in Appendix A.3.

**Pitch prediction.** We train the pitch predictor to predict the per-speaker normalised pitch contour $\widetilde{\mathbf{z}}_{F0}$, from the HuBERT units $\mathbf{z}_c$. The pitch values are normalised to zero mean and standard deviation of one per-speaker on the vocalised sections. We use these values as labels to a predictor conditioned on content units $\mathbf{z}_c$, positional encoding and a speaker vector $\mathbf{z}_{spk}$. The positional encoding is added to $\mathbf{z}_{spk}$ and denotes the absolute position of the unit compared to the start and finish of the sequence, though preliminary results showed that any standard positional encoding works. Intuitively, positional encoding will help learning localised speaker pitch patterns such as ending sentences in a pitch drop.

Our pitch predictor is denoted $E_{F_0}$. An overview of it is described in Figure 3. It is a CNN which has two predicted outputs: binary speaking detection, and a regression head to predict the exact pitch. The binary speaking classification is trained with BCE loss on the binarised pitch labels - i.e. $\mathbf{z}_{F0} > 0$. The regression head is trained with MSE loss on the non-zero, vocalised pitch labels. Like the rhythm predictor this model is trained with a small unit receptive field and unit masking.

### 3.3 Speech Synthesis

We follow Polyak et al. (2021); Kreuk et al. (2021) using a variation of the HiFi-GAN vocoder (Kong et al., 2020). Like previous work, we adapt the generator to take as input a sequence of phonetic-content units, pitch contour and a speaker-embedding. During training, these are only features from real samples as the vocoder is trained independently for reconstruction. At inference time we use the unit sequence inflated with predicted durations, the predicted pitch contour and the target speaker-

embedding. The above features are concatenated along the temporal axis and fed into a sequence of convolutional layers that output a 1-dimensional audio signal. We match the sample rates of unit sequence and F0 by means of linear interpolation, while replicating the speaker embedding.

To sum up, after each of the components described in previous sections are trained, the full pipeline of DISSC is composed of: first extracting content representation, $\mathbf{z}_c$, then, predicting the prosodic features of the target speaker based on $\mathbf{z}_c$ and $\mathbf{z}_{spk}$, and lastly, synthesising the waveform using these new representations.

## 4  SSC Evaluation Suite

In this section we describe our setup which uses several datasets and fine-grained offline metrics for each part of the conversion - timbre, pitch, rhythm. As no standardised framework for evaluating SSC currently exists, we hope the proposed metrics, will help further advance research in this field. A detailed description of training configurations and hyper-parameters can be found in the Appendix.

### 4.1  Datasets

Most previous work focus on a single dataset such as VCTK (Veaux et al., 2017), since few large, multi-speaker datasets with parallel test sets exist. However, VCTK has fairly monotonous speech with less distinct speaking style.

To address this, we use several datasets. First, we use VCTK as it has many speakers. We leave all paired utterances (speakers say the same content) to the test set, to mimic a truly unpaired setup. In preliminary results, having paired data in training improved the results notably, but harms the wanted setup. We follow Qian et al. (2021) in selecting the two fastest (P231, P239) and two slowest (P245, P270) speakers by mean log duration.

In addition, we wished to add a more expressive dataset, yet found that most expressive datasets are from the emotional speech domain. In order to adapt them to our use case we take the Emotional Speech Dataset (ESD) (Zhou et al., 2022), and select one emotion for each of the 10 speakers. This means that all recordings of a given speaker are in a certain emotion thus making it part of their speaking style. Here, we also take the two fastest and two slowest speakers for rhythm metric evaluation.

After preliminary experiments we found that the speakers in the above datasets do not always have clear pitch patterns. This means that the speech is quite monotonous and that variation across recordings was greater than across speakers. To effectively evaluate pitch conversion we created a synthetic dataset based on VCTK - denoted as Syn_VCTK. In this dataset we took 6 speakers from VCTK, encoded them based on the pipeline by Polyak et al. (2021). We then altered the pitch contour by introducing a linear trend of either up, down or flat for each of the speakers. We then generated the new synthetic recordings in the same approach. We encourage readers to listen to these samples as well as the other datasets in the accompanying page to grasp the different speaking styles.

### 4.2  Metrics

#### 4.2.1  Objective Metrics

We introduce new metrics which aim to capture the rhythm in a fine-grained manner, and objectively evaluate the pitch contour (even when the rhythm does not match). We hope these metrics will contribute to other advances in this field.

**Timbre.**  For regular VC we use the common Equal Error Rate (EER) of speaker verification using SpeechBrain's (Ravanelli et al., 2021) ECAPA-TDNN (Desplanques et al., 2020) which achieves 0.8% EER on Voxceleb. For each sample, we randomly select $n$ positive and $n$ negative samples. Positive samples are random utterances of other speakers converted to the current speaker, and negative samples are random utterances of the current speaker converted to a random target speaker.

**Content.**  For phonetic content preservation we use the standard Word Error Rate (WER) and Character Error Rate (CER), using Whisper (Radford et al., 2022) as our Automatic Speech Recognition (ASR) model.

**Rhythm.**  In order to evaluate the rhythm we wish to go beyond the length of the recording, which only indicates the average rhythm and not which sounds are uttered slowly and which quickly. We therefore use the text transcriptions of the samples, and the Montreal Forced Aligner (MFA) (McAuliffe et al., 2017) to get start and end times for each word and phoneme in the recording. We then compare the lengths of the converted speech and the target speech, thus defining Phoneme Length Error (PLE), Word Length Error (WLE) and Total Length Error (TLE). We do not compare

| | MODEL | CONTENT | | RHYTHM&F0 | RHYTHM | | | SPEAKER |
|---|---|---|---|---|---|---|---|---|
| | | WER↓ | CER↓ | EMD↓ | TLE↓ | WLE↓ | PLE↓ | EER↓ |
| VCTK | AutoVC (Qian et al., 2019) | 71.3 | 47.1 | 17.68 | 1.214 | 0.072 | 0.028 | 7.5 |
| | AutoPST (Qian et al., 2021) | 40.6 | 26.7 | 21.9 | 1.379 | 0.123 | 0.037 | 24.1 |
| | Seq2seq-VC (Liu et al., 2021) | **2.9** | **1.2** | 20.95 | 1.214 | 0.072 | 0.028 | 10.9 |
| | SR (Polyak et al., 2021) | 6.6 | 3.3 | 14.0 | 1.214 | 0.071 | 0.025 | 1.8 |
| | DISSC_Rhythm (Ours) | 10.9 | 5.4 | **10.58** | **0.832** | **0.056** | **0.023** | **1.7** |
| | DISSC_Both (Ours) | 13.0 | 6.9 | **10.53** | **0.832** | **0.056** | **0.023** | **1.7** |
| ESD | AutoVC (Qian et al., 2019) | 87.0 | 59.9 | 31.82 | 0.591 | 0.106 | 0.048 | 6.6 |
| | AutoPST (Qian et al., 2021) | 50.3 | 31.8 | 37.2 | 0.549 | 0.097 | 0.043 | 15.7 |
| | SR (Polyak et al., 2021) | **14.9** | **6.0** | 30.3 | 0.591 | 0.097 | 0.041 | 2.9 |
| | DISSC_Rhythm (Ours) | 19.1 | 7.9 | **24.8** | **0.350** | **0.076** | **0.037** | **2.6** |

Table 1: A comparison of the proposed method against VC and SSC baselines considering content (WER, CER), Rhythm and F0 (EMD, TLE, WLE, PLE), and speaker identification (EER). Only DISSC improves rhythm metrics compared to standard VC baseline Speech Resynthesis.

the length of silences as their occurrence can differ between recordings thus breaking the alignment.

Formally, for each input recording and matching text, MFA returns a list of tuples $T_w = ((s_{w_1}, e_{w_1}, c_{w_1}) \ldots (s_{w_m}, e_{w_m}, c_{w_m}))$ for word level and another for phoneme level - $T_p = ((s_{p_1}, e_{p_1}, c_{p_1}) \ldots (s_{p_m}, e_{p_m}, c_{p_m}))$. $s_{\{w,p\}_i}$ indicates the start time of the sound at index $i$ in the recording (word or phoneme according to the level), and $e_{\{w,p\}_i}$ indicates the end time. $c_{\{w,p\}_i}$ represents the content of the time segment, i.e the word or phoneme. We filter only non-silence time segments leaving $\widetilde{T}_{\{w,p\}} = ((s_{\{w,p\}_i}, e_{\{w,p\}_i}, c_{\{w,p\}_i}) \quad \forall i \quad s.t. \quad c_i^{\{w,p\}} \neq silence)$. Thus formally, the errors per sample compare durations of a reference sample and a synthesised sample:

$$
\text{PLE}(\widetilde{T}_p^{syn}, \widetilde{T}_p^{ref}) =
$$
$$
\sum_{j=1}^{len(\widetilde{T}_p)} |(e_{p_j}^{syn} - s_{p_j}^{syn}) - (e_{p_j}^{ref} - s_{p_j}^{ref})|,
$$
$$
\text{WLE}(\widetilde{T}_w^{syn}, \widetilde{T}_w^{ref}) =
$$
$$
\sum_{j=1}^{len(\widetilde{T}_w)} |(e_{w_j}^{syn} - s_{w_j}^{syn}) - (e_{w_j}^{ref} - s_{w_j}^{ref})|.
$$
(1)

The TLE is just the duration difference between the entire recordings. All rhythm metrics are in time units, in this case seconds. In addition, on rare occasions the number of phonemes for the same sound by different speakers is different (due to accents etc.), we ignore these samples for a more well-defined metric. In addition, when the phonetic

content is harmed dramatically in the conversion, MFA might fail to align the text. We penalise such samples by assuming a linear and even split of the recording to phonemes and words.

**Pitch.** Popular evaluation methods for F0 similarity include the Voicing Decision Error (VDE) and F0 frame error (FFE). Intuitively, VDE measures the portion of frames with voicing decision error, while FFE measures the percentage of frames that contain a deviation of more than 20% in pitch value or have a voicing decision error. Formally,

$$
\text{VDE}(\boldsymbol{v}, \hat{\boldsymbol{v}}) = \frac{\sum_{t=1}^{T-1} \mathbb{1}[\boldsymbol{v}_t \neq \hat{\boldsymbol{v}}_t]}{T}, \qquad (2)
$$

$$
\text{FFE}(\boldsymbol{p}, \hat{\boldsymbol{p}}, \boldsymbol{v}, \hat{\boldsymbol{v}}) = \text{VDE}(\boldsymbol{v}, \hat{\boldsymbol{v}})
$$
$$
+ \frac{\sum_{t=1}^{T-1} \mathbb{1}[|\boldsymbol{p}_t - \hat{\boldsymbol{p}}_t| > 0.2 \cdot \boldsymbol{p}_t] \mathbb{1}[\boldsymbol{v}_t] \mathbb{1}[\hat{\boldsymbol{v}}_t]}{T},
$$
(3)

where $\boldsymbol{p}, \hat{\boldsymbol{p}}$ are the pitch frames from the target and generated signals, $\boldsymbol{v}, \hat{\boldsymbol{v}}$ are the respective voicing decisions, and $\mathbb{1}$ is the indicator function.

Although these metrics are widely used in speech synthesis (Wang et al., 2021), these methods are not well adapted to misaligned samples, i.e when they are spoken at different rates. These metrics require the samples to have the same time length, therefore many implementations linearly interpolate the generated sample's pitch contour to that of the target before computing them. However, rhythm differences are non-linear (i.e certain phonemes could be "stretched" more than others) thus not alleviating the misalignment problem. Instead, we suggest using segments obtained from the

forced-aligner and calculate the FFE while interpolating the segments to match the segment sizes. This measures the match in shape and value of the pitch contour for each word and phoneme, even if they are misaligned. We denoted these as P_FFE for phoneme segments and W_FFE for words.

**Rhythm & Pitch.** We compute Earth Movers Distance (EMD) (Rubner et al., 1998) between the pitch contour of converted and target samples. Essentially we check how much "pitch mass" needs to be moved between them. This metric considers errors along the time domain (rhythm) and pitch contour errors (such as predicting the wrong trend). Our results show strong correlation to human perception of rhythm and pitch with the above metrics.

This newly proposed evaluation suite is generic and can transfer to other languages. However, certain metrics are based on language-specific pre-trained models (e.g., ASR-based metrics). In such cases, a multi-lingual ASR can be used. In our setup, we use the Whisper model (Radford et al., 2022) which supports $\sim 100$ languages. Another option would be to use the MMS model (Pratap et al., 2023) which supports more than 1000 languages for both ASR and forced-alignment.

#### 4.2.2 Human evaluation

To confirm that the proposed objective metrics correlate to human perception, and to evaluate to what extent the prosody impacts humans' speaker recognition we conduct a human evaluation study. We use the common mean opinion score (MOS) to evaluate the quality and naturalness of generated and real samples. Specifically, each sample was scored from 1-5 on naturalness and quality. We took the same 40 samples for each method and the original. Each sample was annotated by at least three raters.

Next, to evaluate the effectiveness of the speaking style conversion, we present the users a sample from the target speaker and a sample converted by each method. Users rated each sample from 1-5 ("Very Different"-"Very Similar") on how likely it is to be uttered by the same speaker and instructed to "pay attention to speaking style such as rhythm and pitch changes, and not only the voice texture". We evaluated 40 samples per-method, with at least three raters per-sample. We report the mean score and 95% confidence intervals for both tests. We modified the WebMUSHRA[1] tool for the tests. A

---

[1] https://github.com/audiolabs/webMUSHRA

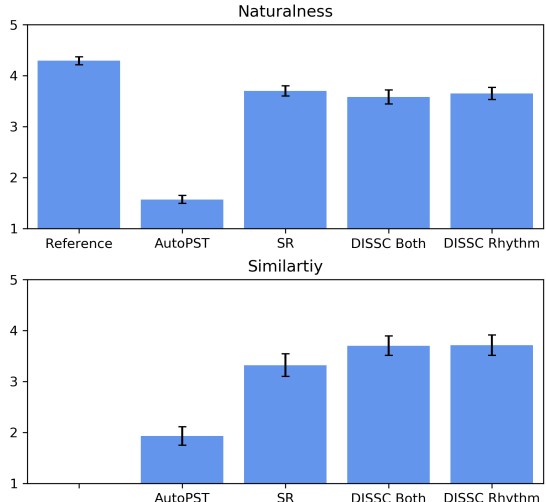

Figure 4: We compare the human evaluated naturalness of real reference utterances, Speech Resynthesis (SR), AutoPST, two versions of DISSC which convert rhythm only or also F0. We also compare the similarity of how likely the converted sample is to come from the the same speaker as the target recording.

detailed description of the subjective tests can be found in the Appendix.

## 5 Results

We compare DISSC to several any-to-many (or many-to-many) baselines. Notably, we evaluate Speech Resynthesis (SR), and AutoVC (Qian et al., 2019) which are well known VC methods. In addition, we compare to AutoPST and BNE-MoL-seq2seqVC (Liu et al., 2021) which are prominent prosodic-aware VC (i.e., SSC) methods. We use the official Github implementations and train the models on our datasets using the authors recommended configuration. For a fair comparison, all baselines were evaluated under the any-to-many setup, i.e., generating only seen speakers. Only seq2seqVC supports new speakers, because only that approach was published. However, we evaluate them on known speakers, with the entire training set as reference recordings. We show three variants of our method: DISSC_Rhythm, DISSC_Pitch and DISSC_Both which predict duration, pitch contour and both respectively.

Objective results for VCTK and ESD appear in Table 1. On both datasets DISSC improves the length errors across all scales: phonemes, words and utterances. The relative improvement compared to the next best baseline is 56.8% on ESD TLE. At word level we get 27.6% and 26.7% rela-

| MODEL | CONTENT | | RHYTHM&F0 | RHYTHM | | | F0 | |
|---|---|---|---|---|---|---|---|---|
| | WER↓ | CER↓ | EMD↓ | TLE↓ | WLE↓ | PLE↓ | W_FFE↓ | P_FFE↓ |
| SR (Polyak et al., 2021) | **12.4** | **6.3** | 24.23 | 1.101 | 0.064 | 0.024 | 46.1 | 43.4 |
| DISSC_Rhythm (Ours) | 18.8 | 9.5 | 20.78 | **0.775** | **0.053** | 0.023 | 46.1 | 43.7 |
| DISSC_Pitch (Ours) | **12.5** | **6.3** | 13.55 | 1.101 | 0.065 | 0.024 | **24.5** | **20.5** |
| DISSC_Both (Ours) | 19.6 | 10.2 | **10.47** | **0.775** | **0.053** | **0.023** | 25.3 | 21.4 |

Table 2: VC and SSC results on the Syn_VCTK dataset. This ablation measures the impact of rhythm and pitch conversion over SR considering content (WER, CER), Rhythm and F0 (EMD, TLE, WLE, PLE, W_FFE, P_FFE).

tive improvement on ESD and VCTK respectively. All other baselines show rhythm errors comparable or worse than SR which does not convert rhythm.

We can see a minor decrease in content quality (by WER and CER) of DISSC compared to SR which does not convert prosody, but still superior to AutoPST. In addition, seq2seq also has better WER, at the expense of harming the rhythm conversion. Potentially, the supervised phoneme recogniser is too fine, thus does not impose enough of a rhythm bottleneck. The harm in quality of DISSC may be due to additional information bottlenecks by de-duplicating the speech units, and predicting F0.

We additionally provide subjective evaluation considering both naturalness of the converted samples and their similarity to the target recording considering speaking style. These results are shown in Figure 4. We found that the naturalness of both DISSC variants was comparable to SR, and not far from the reference recordings while far outperforming AutoPST. In addition, we see a noticeable increase in speaking style similarity when converting prosody with DISSC compared to SR. However, also converting the pitch did not noticeably improve the results. We believe that this has to do with the lack of distinct speaking style which people can notice, especially from a single recording.

**Ablation study.** In order to properly evaluate the ability of DISSC to learn the pitch speaking style, when such exists, we use the synthetic VCTK data (as explained in Section 4.1) with known pitch trends. These results (Table 2) are an ablation study for each converter of DISSC. They highlight the control-ability of our approach - when predicting the rhythm, only length errors improve, likewise for pitch. We see that when a pitch pattern exists in the speaking style - DISSC learns it, providing 88-113% relative increase in the metrics. We also see that they can be converted jointly without noticeable drop in pitch and length performance.

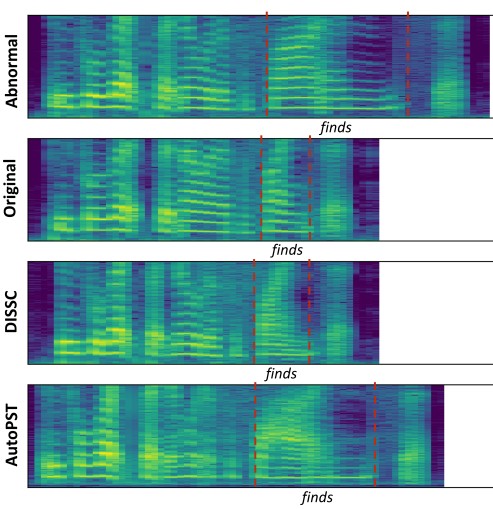

Figure 5: DISSC corrects abnormal rhythm patterns in specific words, compared to AutoPST. Here we take the utterance "people look, but no one ever finds it" and stretch the duration of the word "finds" times three.

**Irregular rhythm.** Lastly, to demonstrate that DISSC predicts rhythm depending on content, and not only on average - we create a sample with abnormal rhythm patterns. We take an utterance and artificially stretch one of the words using speech resynthesis. We then attempt to "convert" the sample to the original speaker thus checking the ability to only change the rhythm of the abnormal part. We compare DISSC and AutoPST in Figure 5, and see that DISSC manages to mostly correct the irregular rhythm, while hardly changing the other parts. By contrast, AutoPST only slightly changes the rhythm. If $E_c$ performs optimally, we are guaranteed perfect results, as $z_c$ would be identical for irregular and original after dedup. However, HuBERT may introduce some errors specifically when considering signal variations (Gat et al., 2022).

## 6 Conclusion & Future Work

In this work, we propose a simple and effective method for *speaking style conversion* based on discrete self supervised speech representations. We

provide a formal definition to the speaking style conversion setup while introducing a set of evaluation functions to analyse and evaluate different speech characteristics separately. Results suggest, the proposed approach is superior to the evaluated baselines considering both objective and subjective metrics. We hope these advance the study in the field, and lead to improved performance.

The current model suffers from slight content loss when using the deduped units. In addition, repeated utterances get slightly different units (in the abnormal rhythm experiment). Therefore, in future work we aim to improve the robustness and disentanglement of the speech representation considering speaker and prosodic information.

## Limitations

Our approach can only perform conversion into a closed set of speakers seen at train time. This is due to the LUT used to learn speaker representations and to the use of per-speaker pitch statistics. This might be hard to overcome as speaker style (rhythm and pitch changes) can be hard to judge from a small number of samples. Additionally, we see that speaking style conversion and especially rhythm conversion harm the WER and CER slightly. The additional bottleneck of de-duplication which allows us to convert rhythm comes with the cost of slight content loss. We hope that future improvement of the hidden representations will help improve or alleviate the tradeoff.

## Ethical Statement

The broader impact of this study is, as in any generative model, the development of a high quality and natural speech synthesis model. This has special sensitivity and ethical concerns as such technology might be used to alter voices and speaking style in speech recordings, which can be considered as biometric data and biometric processing. To deal with that we limit the number of speakers that can be synthesized with the proposed approach using a pre-defined voices using a look-up-table. Another potential risk of the proposed method which is also one of its limitations is that the generated speech content is not always perfect, hence might lead to wrong pronunciations.

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

# A Appendix

Here we give fine-grained details of computational costs, hyper-parameters and other implementation details omitted from the main part for brevity and readability.

## A.1 Dataset details

We use several datasets: (i) VCTK dataset (Veaux et al., 2017); (ii) ESD (Zhou et al., 2022); (iii) and the a synthetic VCTK dataset (Syn_VCTK), generated by us. For VCTK the training data contains 41k samples, and the evaluation set contains 2.5k. For ESD the training data contains 3k samples, the validation contains 200, and the test 300. For Syn_VCTK the training data contains 2.4k samples and the validation has 142. For ESD we use the official train, val, test split. For both VCTK versions we enforce unpaired data in the train sets, and use paired data only for evaluation.

## A.2 Computational Price

We use a pretrained HuBERT_Base model for input encoding. This model has 90 million parameters and takes less than 15 minutes to inference over all of our datasets. For training times see the original paper. For the HiFi-GAN vocoder which is trained once, independently of the speaking style conversion, we use a model with 13.8 million parameters (for the generator). It takes 4 days on to train on VCTK using two RTX 6000 GPUs, other datasets take less than that.

The rhythm predictor $E_{dur}$ has 330 thousand parameters, and takes about 30 minutes to train on a single RTX2080 GPU. Likewise, the pitch predictor $E_{f0}$ has 527 thousand parameters, and also takes about 30 minutes to train on a single RTX2080 GPU.

## A.3 Hyper-Parameters

We encode the HuBERT units and pitch contour with *textlesslib*. For the content endocer we use *hubert-base-ls960* and use K-means with 100 clusters for the clustering step. For the pitch contour we use the default parameters from textless lib which match the content encoder we used.

For the vocoder we use the official implementation of speech resynthesis with the configuration suitable for VCTK, hubert100, using a fixed size lookup-table. We adapt the model so that it receives as input the normalised pitch without quantisation. We train it with a batch size of 64 over two

GPUs for the minimum between 2000 epochs and 400,000 batches.

The rhythm predictor $E_{dur}$ is built of 8 blocks of Conv1D followed by Batch Normalisation, with 128 hidden units. The speaker embedding and token embedding are learned with a lookup-table, both with size 32. The speaker vector is repeated along the time axis and concatenated with the unit embedding. Each CNN has a receptive field of 3 and padding of 1 so that the output shape is the same as the input. For activation we use Leaky ReLU. We normalise the length for prediction with the mean and standard deviation of the training data. Train time masking masks each unit in the input independently with 20% probability. We train the model for 30 epochs, with a batch size of 32, a learning rate of $3e-4$ and Adam optimiser. We take the model with lowest validation MSE. This architecture and parameters were fairly standard values which worked out of the box, no manual or automated hyper-tuning was done.

The pitch predictor $E_{F0}$ encodes the speaker and units in the same way as $E_{dur}$ with a fixed size lookup-table of size 32. However, here we add to the speaker embedding a linear positional encoding - this means that indices 0-15 of the speaker vector get added the unit index relative to the start and indices 16-31 get added the unit index relative to the end. In preliminary experiments more typical sinusoidal positional encoding worked similarly but we chose the linear version for simplicity. The model is comprised of 9 shared Conv1D layers with kernel size 3, and followed by 2 Conv1D with kernel sizes 3 and 1 for each head (speaking classification and F0 regression). We use Leaky ReLU activation. The labels for training are received from the YAAPT algorithm, normalised per-speaker with the mean and standard deviation of the vocalised parts of the training set. We do not normalise the non-vocalised parts. The BCE loss for binary classification is multiplied by 100 and added to the pitch MSE loss, to attempt to give both losses a similar scale. We train the model for 20 epochs with learning rate $3e-4$, batch size of 32 and Adam optimiser. We take the model with the lowest validation MSE. The results in Table 1, and Figure 4 are with a similar model, however it does not use positional encoding. The positional encoding was added afterwards as an improvement. This architecture and parameters were fairly standard values which worked out of the box, no manual or automated hyper-tuning was done.

### A.4 Human Evaluation Details

We had over 20 different annotators in our subjective tests, who were all non-paid volunteers. They are all fluent, though not necessarily native, in English, and vary in academic background and connection to the field. For privacy reasons the questionnaires are anonymous and no details are collected about the participants.

The quality question is phrased as follows: *"Evaluate the quality and naturalness of the following audio segment"*, and the raters give a scale of 1-5 with the labels: *"Poor"*, *"Decent"*, *"Good"*, *"Very Good"*, *"Excellent"*.

The similarity question is phrased as follows: *"The first recording in this section is a reference, mark all recordings in how similar the speaking style is to the original. speaking style refers to the speaking rhythm, intonation etc. Try to ignore the quality of the recording itself"*, and the raters give a scale of 1-5 with the labels: *"Very different"*, *"Different"*, *"Slightly similar"*, *"Similar"*, *"Very Similar"*.

### A.5 Low Resource Language Conversion

As mentioned, DISSC does not use text as any intermediate representation in conversion (or text supervision for training). This means that it can work in low-resource languages which do not have large annotated datasets.

We demonstrate the ability of DISSC to convert a Hebrew utterance to a target VCTK speaker, while not being trained on any Hebrew data (see Fig. 6). While converting the accent in cross-lingual (Hebrew utterance to English speaker) remains challenging, this shows good potential. Notably, this example is out of domain for all model training data. Training DISSC in a *textless self-supervised* way on Hebrew data will likely improve the performance dramatically.

### A.6 WER detailed analysis

While DISSCis the only evaluated method to successfully convert prosody, it has a limitation of slightly harming content compared to some methods which do not convert prosody (e.g. speech resythesis). We see this in the WER and CER metrics and wish to further analyse the cause for this.

We first manually inspect the samples with highest word error rate. We have found that the content loss is mainly due to minor phonological errors

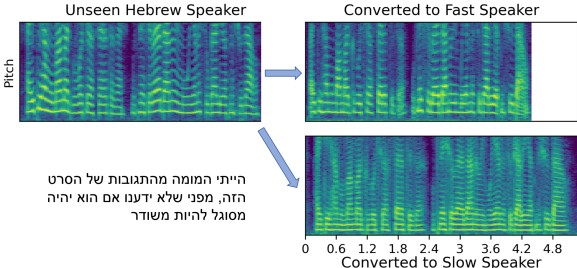

Figure 6: An example using DISSC to convert a non-English sample. Hebrew is a lower resource language, hence there are little to no transcribed datasets. Thus, no expressive TTS methods exist, making an ASR-TTS VC pipeline impossible. Training DISSC on *unlabelled* Hebrew data, will likely improve results further. We encourage readers to listen to the accompanying samples.

| Source Speaker | Method | |
|---|---|---|
| | SR | DISSC_Rhythm |
| p231 | 6.4 | 11.1 |
| p239 | 12.2 | 18.6 |
| p245 | 4.5 | 7.9 |
| p257 | 9.8 | 15.7 |
| p270 | 3.0 | 7.7 |
| p284 | 12.1 | 18.2 |
| p306 | 4.5 | 7.0 |
| p341 | 1.3 | 4.0 |

Table 3: Word error rate comparison on VCTK, split by source speaker when converting to all other speakers as targets. For instance the first row denotes the WER when converting a fixed set of samples from speaker p231 to all other speakers in the list. We compare DISSC_Rhythm and Speech resynthesis and see that the results vary greatly, and that the success of both methods are correlated.

which cause transcription change. For example, in cases with a high error rate, the original text is: "Please call Stella" and the transcription after conversion is: "Peace cool Stalla". However, when looking at the WER of both DISSC and speech resynthesis, we observe that such phonological errors are observed in the same samples for both methods (even without conversion). Therefore, we believe that the root cause of such phenomenon is the quality of the SSL model and DISSC amplified this effect.

To further, evalute this hypothesis we evaluate the WER per source speaker and per target speaker (results in Tables 3 and 4). This analysis clearly shows that WER per-speaker varies greatly based on the source speaker, but does not depend on the

| Target Speaker | Method | |
|---|---|---|
| | SR | DISSC_Rhythm |
| p231 | 6.2 | 11.2 |
| p239 | 7.0 | 11.4 |
| p245 | 6.6 | 11.6 |
| p257 | 6.4 | 10.5 |
| p270 | 7.2 | 10.9 |
| p284 | 6.4 | 11.5 |
| p306 | 7.2 | 11.4 |
| p341 | 7.1 | 11.8 |

Table 4: Word error rate comparison on VCTK, split by target speaker when converting from all other speakers as sources. For instance the first row denotes the WER when converting a fixed set of samples from all source speakers in the list (except p231) to speaker p231. We compare DISSC_Rhythm and Speech resynthesis and see that the results hardly depend on the target speaker.

target speaker, this further supporting the fact that this is an issue with the SSL encoding method. An interesting line of future work would be in improving SSL for such cases, or even applying error-correcting codes on the converted speech.