# OpenReview forum: "Speaking Style Conversion in the Waveform Domain Using Discrete Self-Supervised Units"
_EMNLP/2023/Conference — EMNLP 2023 Findings_

### Official Review · Reviewer_GxPi · 2023-08-04

**Soundness:** 4

**Excitement:**

4: Strong: This paper deepens the understanding of some phenomenon or lowers the barriers to an existing research direction.

**Paper Topic And Main Contributions:**

This paper addresses the problem of speaking style conversion (SSC) in the field of natural language processing (NLP). Speaking style conversion involves transforming the prosodic features of a speech signal, such as pitch and rhythm, while preserving the phonetic content and speaker identity. The main contribution of this paper is the introduction of a novel approach called "DISSC" (Discrete Self-Supervised Speech Conversion) for speaking style conversion using discrete self-supervised speech representations, the introduction of new evaluation metrics, and the demonstration of superior performance compared to existing baselines. These contributions advance the research in the field of speech synthesis and have potential applications in various NLP-related tasks.

**Questions For The Authors:**

1. Can you provide more details about the generalization capabilities of DISSC? Specifically, how does the model perform on unseen speakers, and what strategies have been employed to address the closed-set limitation?

2. The paper mentions slight content loss during de-duplication. Can you elaborate on the impact of this content loss on the overall quality of the generated speech? Are there any strategies or techniques considered to mitigate this issue?

3. Can you provide more insights into the failure cases of DISSC? An error analysis highlighting the limitations and potential weaknesses of the proposed method would be valuable for understanding its performance and future improvements.

**Reasons To Accept:**

1. Approach: The paper proposes a novel approach, DISSC, for speaking style conversion using discrete self-supervised speech representations. This novel method contributes to the advancement of speech synthesis techniques and provides an innovative solution to the speaking style conversion problem.

2. Formal Definition and Evaluation Metrics: The paper provides a formal definition of the speaking style conversion setup, which adds clarity and structure to the research in this area. Moreover, the introduction of new fine-grained offline evaluation metrics, such as Phoneme Length Error, Word Length Error, Total Length Error, and Earth Movers Distance (EMD) for pitch contour comparison, helps in assessing the quality of speaking style conversion in a more detailed manner, which can be valuable for future research and benchmarking.

3. Empirical Results: The paper presents comprehensive empirical results that demonstrate the superiority of DISSC over several baseline methods, including both objective and subjective evaluations. This empirical evidence and performance comparison provide confidence in the effectiveness of the proposed approach and its potential practical applicability.

4. Ablation Study and Control-ability: The inclusion of an ablation study that analyzes the effect of individual components of DISSC shows the control-ability of the approach in handling rhythm and pitch conversion separately or jointly. This understanding of the model's behavior is valuable for researchers and developers seeking to tailor the approach to specific requirements.

**Reasons To Reject:**

1. Content Loss in De-Duplication: The paper mentions that de-duplication of speech units may lead to slight content loss. This content loss might impact the overall quality of the generated speech, and the paper does not provide a detailed analysis of this effect or a thorough discussion on mitigating this issue.

2. Lack of Large-Scale Experiments: The paper presents results on datasets with a limited number of speakers (e.g., VCTK and ESD). Conducting experiments on larger and more diverse datasets, encompassing a broader range of speaking styles, would strengthen the validity of the findings and provide more robust insights into the performance of DISSC across different scenarios.

3. Limited Generalization: The proposed method appears to be limited to a closed set of speakers seen during training. This restricts the applicability of DISSC to unseen speakers, which is a common requirement in real-world scenarios. Addressing this limitation and demonstrating the ability to generalize to unseen speakers would significantly enhance the practical utility of the proposed approach.

**Reproducibility:**

4: Could mostly reproduce the results, but there may be some variation because of sample variance or minor variations in their interpretation of the protocol or method.

**Reviewer Confidence:**

3: Pretty sure, but there's a chance I missed something. Although I have a good feel for this area in general, I did not carefully check the paper's details, e.g., the math, experimental design, or novelty.

---

> ### Author Rebuttal · Authors · 2023-08-27
>
> We highly appreciate the reviewer’s feedback, and their comments and suggestions for future work are of great interest. We are delighted that __they found our approach novel and acknowledged the value of the evaluation setup__. We are pleased that they found the __evaluation and ablation studies to be comprehensive__.
>
> __Regarding the content de-duplication loss__ - As mentioned in the paper (lines 595-598, 609-614), the de-duplication and the ability of the HuBERT units to model content independently are not perfect. We analyze the impact of such phenomena through the ASR metric when de-duplicating (converting rhythm) compared to only converting pitch in Table 1. As we discussed in Section 6, there could be different methods to improve this including better SSL models (with higher correlation to phonemes), use of more expressive continuous representation for each de-duplicated time step, etc. We believe these will be addressed in future work.
> However, when considering the quality of the converted speech (i.e., MOS scores), DISSC still outperforms the SSC evaluated baselines and comparable to the baselines which do not alter the prosodic features (i.e., speech-resynthesis [2]).
>
> Nevertheless, following the reviewer’s comment, we further analyze conversions with respect to the content loss, see the Tables below and paragraph below.
>
> __Regarding the lack of large scale experiments__ – We follow the standard practice in the literature [1,2,3], and even go beyond these by adding ESD and synthetic VCTK. We hope that formalizing the setup, including the evaluation requirements, will drive data annotation efforts which will create even larger real-world datasets for speaking style conversion.
>
> __Regarding generating only seen speakers__ - Speaking style tends to be more diverse than speaker timbre, we believe it is harder to capture true speaking style from only a few seconds. Hence, we focus on seen speakers only. Moreover, as stated in the paper (lines 208-211), prior work found that limiting the generator to seen speakers only, improves the generation quality [2]. Furthermore, while some works convert prosodic features based on a single utterance [6] this does not capture the general speaking style of the speaker.
>
> __Regarding error analysis__ - Thanks for the suggestion, following the reviewer’s request we conducted a fine-grained analysis to better understand the limitations of the proposal with respect to content loss. We analyze the WER as a function of the different source speakers and target speakers. See the Tables below.
>
> | source speaker \ WER | Speech-Resynthesis | DISSC_Rhythm |
> |----------------------|----|--------------|
> | p231                 |  6.4  | 11.1         |
> | p239                 |  12.2  | 18.6         |
> | p245                 |  4.5  | 7.9          |
> | p257                 |  9.8  | 15.7         |
> | p270                 |  3.0  | 7.7          |
> | p284                 |  12.1  | 18.2         |
> | p306                 |  4.5 | 7.0          |
> | p341                 |  1.3  | 4.0          |
>
>
> | target speaker \ WER | Speech-Resynthesis | DISSC_Rhythm |
> |----------------------|----|--------------|
> | p231                 |  6.2  | 11.2         |
> | p239                 |  7.0  | 11.4         |
> | p245                 |  6.6  | 11.6         |
> | p257                 |  6.4  | 10.5         |
> | p270                 |  7.2  | 10.9         |
> | p284                 |  6.4  | 11.5         |
> | p306                 |  7.2  | 11.4         |
> | p341                 |  7.1  | 11.8          |
>
> We have found the content loss is mainly due to minor phonological errors which cause transcription change. For example, in cases with a high error rate, the original text is: “Please call Stella” and the transcription after conversion is: “Peace cool Stalla”.
> However, when looking at the WER of both DISSC and speech resynthesis, we observe the phonological errors are also observed in the same samples in both methods (even without conversion), hence, we believe the root cause of such phenomenon is in the quality of the SSL model and DISSC amplified such an effect. The fact that WER per-speaker varies greatly based on the source speaker, and not the target speaker further supports the fact that this is an issue with the SSL encoding method. An interesting line of future work would be in improving SSL for such cases, or even applying error-correcting codes on the converted speech.
>
> &nbsp;
>
> [1] Kaizhi Qian, Yang Zhang, Shiyu Chang, Jinjun Xiong, Chuang Gan, David Cox, and Mark Hasegawa-Johnson. 2021. Global prosody style transfer without text transcriptions. In International Conference on Machine Learning, pages 8650–8660. PMLR.
>
> [2] Adam Polyak, Yossi Adi, Jade Copet, Eugene Kharitonov, Kushal Lakhotia, Wei-Ning Hsu, Ab-delrahman Mohamed, and Emmanuel Dupoux. 2021. Speech resynthesis from discrete disentangled self-supervised representations. arXiv preprint arXiv:2104.00355
>
> [3] Songxiang Liu, Yuewen Cao, Disong Wang, Xixin Wu, Xunying Liu, and Helen Meng. 2021. Any-to-many voice conversion with location-relative sequence-to-sequence modeling. IEEE/ACM Transactions on Audio, Speech, and Language Processing, 29:1717–1728.
>
> [6] Kaizhi Qian, Yang Zhang, Shiyu Chang, Mark Hasegawa-Johnson, and David Cox. 2020. Unsupervised speech decomposition via triple information bottleneck. In International Conference on Machine Learning, pages 7836-7846. PMLR.

---

### Official Review · Reviewer_RPWj · 2023-08-04

**Soundness:** 4

**Excitement:**

4: Strong: This paper deepens the understanding of some phenomenon or lowers the barriers to an existing research direction.

**Paper Topic And Main Contributions:**

This paper is focusing on one of the hot topic in TTS- speaker conversion. It talks about the most difficult part of this task- conversion of the speaker style. While there are a lot of methods that allow to make voice identity conversion, speaker style is non trivial. The author propose simple and effective method for speaker style conversion, moreover they also propose a way how to measure the quality of style conversion.

**Reasons To Accept:**

I think this paper is a good candidate to be accepted because of its 1) relevance, 2) novelty 3) complexity of evaluations performed. This paper will be interesting for a wide audience of EMNLP, both for academia and industry professionals. The topic of voice conversion is very popular especially in industry application where the style transfer of the voice is the most complicated task. The paper provides a novel approach for style transfer and demonstrates through a thorough evaluation that is superior against the baseline.  On top of it they provide an ablation study that gives extra insights into the model. As a final important reason to accept this paper I consider a proposal of a complex and diverse approach on how to evaluate the style transfer between voices, which a non trivial task with limited literature on it.

**Reasons To Reject:**

There is not much attention in this paper on how this method on both voice style conversion and its evaluation transfers across languages and what are the main limitations expected.

**Reproducibility:**

4: Could mostly reproduce the results, but there may be some variation because of sample variance or minor variations in their interpretation of the protocol or method.

**Reviewer Confidence:**

4: Quite sure. I tried to check the important points carefully. It's unlikely, though conceivable, that I missed something that should affect my ratings.

---

> ### Author Rebuttal · Authors · 2023-08-27
>
> We would like to thank the reviewer for taking the time to carefully read our paper and provide valuable feedback. We are happy to hear the reviewer __found our paper relevant, novel and thoroughly evaluated__. We are also glad that they recognised the __importance of the evaluation suite proposed for future research__.
>
>
> __Regarding the lack of attention to other languages__ – we agree that developing speaking style conversion (SSC) models for more languages is an interesting line of research. Unfortunately, in that setting, there is a lack of large datasets with parallel, expressive speech. Moreover, to be consistent with the literature we follow prior work which focuses on English datasets [1, 2, 3]. However, unlike many existing methods, our approach is textless which means datasets without text annotations are also relevant for training. Hence, “lowering the bar” for SSC in other languages.
> We demonstrate in the appendix the ability to perform SSC on a low-resource language without additional training in the target language (i.e., zero-shot language transfer). This demonstrates the potential for low-resource languages when performing textless self-supervised training, and hopefully opens the door to multilingual models in future research.
>
> __Regarding the ability of the evaluation methods to transfer to different languages__ – the newly proposed evaluation suite is generic and can transfer to other languages. However, certain metrics are based on language-specific pre-trained models (i.e., ASR-based metrics). In such cases, a multi-lingual ASR can be used. In our setup, we use the Whisper model [4] which does support ~100 languages. Another option would be to use the MMS model [5] which supports more than 1000 languages for both ASR and forced-alignment.
>
> &nbsp;
>
>
> [1] Kaizhi Qian, Yang Zhang, Shiyu Chang, Jinjun Xiong, Chuang Gan, David Cox, and Mark Hasegawa-Johnson. 2021. Global prosody style transfer without text transcriptions. In International Conference on Machine Learning, pages 8650–8660. PMLR.
>
> [2] Adam Polyak, Yossi Adi, Jade Copet, Eugene Kharitonov, Kushal Lakhotia, Wei-Ning Hsu, Ab-delrahman Mohamed, and Emmanuel Dupoux. 2021. Speech resynthesis from discrete disentangled self-supervised representations. arXiv preprint arXiv:2104.00355
>
> [3] Songxiang Liu, Yuewen Cao, Disong Wang, Xixin Wu, Xunying Liu, and Helen Meng. 2021. Any-to-many voice conversion with location-relative sequence-to-sequence modeling. IEEE/ACM Transactions on Audio, Speech, and Language Processing, 29:1717–1728.
>
> [4] Radford, Alec, et al. "Robust speech recognition via large-scale weak supervision." International Conference on Machine Learning. PMLR, 2023.
>
> [5] Pratap, Vineel, et al. "Scaling speech technology to 1,000+ languages." arXiv preprint arXiv:2305.13516 (2023).

---

### Official Review · Reviewer_rc2g · 2023-08-06

**Soundness:** 1

**Excitement:**

1: Poor: I cannot identify the contributions of this paper, or I believe the claims are not sufficiently backed up by evidence. I would fight to have it rejected.

**Paper Topic And Main Contributions:**

I will reject this paper due to a violation of blind review, as I discovered that it was publicly available at https://arxiv.org/pdf/2212.09730.pdf.

**Reasons To Accept:**

I will reject this paper due to a violation of blind review, as I discovered that it was publicly available at https://arxiv.org/pdf/2212.09730.pdf.

**Reasons To Reject:**

I will reject this paper due to a violation of blind review, as I discovered that it was publicly available at https://arxiv.org/pdf/2212.09730.pdf.

**Reproducibility:**

N/A: Doesn't apply, since the paper does not include empirical results.

**Reviewer Confidence:**

5: Positive that my evaluation is correct. I read the paper very carefully and I am very familiar with related work.

---

> ### Author Rebuttal · Authors · 2023-08-27
>
> We believe there is a misunderstanding as __we do not violate the double-blind review policy__. While we do have a non-anonymous pre-print published online this is in line with the official EMNLP anonymity guidelines. As the official guidelines state _“If you have posted a non-anonymized version of your paper online before the start of the anonymity period, you may submit an anonymized version to the conference. The submitted version must not refer to the non-anonymized version, and you must inform the program chairs that a non-anonymized version exists”_. We uploaded the non-anonymous version much before the start of the anonymity period (uploaded in December 2022, while anonymity starts in May 2023). We would also like to highlight that we did not promote it online or make changes during the anonymity period.
>
> We hope the reviewer reconsiders their review and will rate our paper based on its merit.

---

### Meta-Review · Area_Chair_ANER · 2023-09-16

**Recommendation:** 4

**Metareview:**

The paper address the problem of speaking style conversion and proposed a approach DISSC.

Pros:
- comprehensive results for comparing with baseline methods
- Showing evaluation criteria for speaking style conversion.
Cons:
- Large scale speaker experiment would be more helpful
- There is some content loss with de-duplication.

---

### Decision · Program_Chairs · 2023-10-07

**Decision:**

Accept-Findings

**Comment:**

The paper address the problem of speaking style conversion and proposed a approach DISSC.

Pros:
- comprehensive results for comparing with baseline methods
- Showing evaluation criteria for speaking style conversion.
Cons:
- Large scale speaker experiment would be more helpful
- There is some content loss with de-duplication.